# Diffusion-Weighted Magnetic Resonance Imaging as a Noninvasive Parameter for Differentiating Benign and Malignant Intraperitoneal Collections

**DOI:** 10.3390/medicina56050217

**Published:** 2020-05-01

**Authors:** Paul-Andrei Ștefan, Csaba Csutak, Andrei Lebovici, Georgeta Mihaela Rusu, Carmen Mihaela Mihu

**Affiliations:** 1Anatomy and Embryology, Morphological Sciences Department, “Iuliu Haţieganu” University of Medicine and Pharmacy Cluj-Napoca, 400012 Cluj-Napoca, Romania; stefan_paul@ymail.com; 2Radiology and Imaging Department, County Emergency Hospital, 400012 Cluj-Napoca, Romania; andrei1079@yahoo.com (A.L.); mihageorgeta@yahoo.com (G.M.R.); carmenmihu2004@yahoo.com (C.M.M.); 3Radiology, Surgical Specialties Department, “Iuliu Haţieganu” University of Medicine and Pharmacy Cluj-Napoca, 400012 Cluj-Napoca, Romania; 4Histology, Morphological Sciences Department, “Iuliu Haţieganu” University of Medicine and Pharmacy Cluj-Napoca, 400012 Cluj-Napoca, Romania

**Keywords:** ascites, diffusion-weighted imaging (DWI), magnetic resonance (MRI), peritoneal carcinomatosis

## Abstract

*Background and Objective*: The imaging differentiation of benign from malignant intraperitoneal collections (IPCs) relies on the tumoral morphological modifications of the peritoneum, which are not always advocating for malignancy. We aimed to assess ascitic fluid with the apparent diffusion coefficient (ADC) to determine non-invasive, stand-alone, differentiation criteria for benign and malignant intraperitoneal effusions. *Materials and Methods*: Sixty-one patients with known IPCs who underwent magnetic resonance examinations for reasons such as tumor staging, undetermined abdominal mass and disease follow up were retrospectively included in this study. All subjects had a final diagnosis of the fluid based on pathological examinations, which were divided into benign (n = 37) and malignant (n = 24) IPCs groups. ADC values were measured separately by two radiologists, and the average values were used for comparing the two groups by consuming the independent samples *t*-test. The receiver operating characteristic analysis was performed to test the ADC values’ diagnostic ability to distinguish malignant from benign collections. *Results*: The differentiation between benign and malignant IPCs based on ADC values was statistically significant (*p* = 0.0034). The mean ADC values were higher for the benign (3.543 × 10^−3^ mm^2^/s) than for the malignant group (3.057 × 10^−3^ mm^2^/s). The optimum ADC cutoff point for the diagnosis of malignant ascites was <3.241 × 10^−3^ mm^2^/s, with a sensitivity of 77.78% and a specificity of 80%. *Conclusions*: ADC represents a noninvasive and reproducible imaging parameter that may help to assess intraperitoneal collections. Although successful in distinguishing malignant from benign IPCs, further research must be conducted in order to certify if the difference in ADC values is a consequence of the physical characteristics of the ascitic fluids or their appurtenance to a certain histopathological group.

## 1. Introduction

Peritoneal carcinomatosis (PC) can have a wide spectrum of imaging appearances, such as malignant ascites, omental involvement, infiltration of the mesentery and serous peritoneal implants [1]. Ultrasound and computer tomography are the most common imaging techniques in the assessment of PC [2,3]. Because these methods mainly rely on morphology, their contribution in differentiating scar tissue from peritoneal implants and the identification of PC without a circumscript tumor may be limited [4]. Such difficulties can be partially solved by contrast-enhanced magnetic resonance, which provides information about tissue vascularization along with high tissue contrast [5]. Although they are better highlighted by magnetic resonance (MRI), the morphological changes caused by PC are inconsistent and many may occur in later stages of the disease [6]. Malignant ascites, however, represent an early sign that can be found in two-thirds of patients with PC [1], even when the primary tumor site remains undetected [7].

The differentiation between benign and malignant peritoneal effusions is currently based on the cytological examination of the fluid extracted by paracentesis or by intra-operative sampling, maneuvers that, in addition to being invasive, also expose the patients to a series of risks [7]. Upon pathological examination, the two types of liquid have their own characteristics in terms of physical, biochemical properties and cellularity [8]. It is desirable that these features are also reflected in medical images, and that their appearance is too subtle to be visually quantified. Such limitations created the need for means of a quantitative assessment of the information comprised in medical images [9]. Diffusion-weighted imaging (DWI) is an MRI technique, which, via the measurement of the apparent diffusion coefficient (ADC), provides tissue analysis based on the diffusion of water molecules [10]. This proves that DWI can reflect tissue properties such as viscosity and cellularity features [10,11], and therefore may be useful in highlighting the characteristics of different types of ascitic fluids. We investigated the ability of ADC measurements to distinguish between benign and malignant intraperitoneal collections (IPCs) in order to obtain a non-invasive additional diagnostic criterion for intraperitoneal effusions.

## 2. Materials and Methods

### 2.1. Study Group

This single-institution retrospective study was approved by the institutional review board (ethics committee of the “Iuliu Hațieganu” University of Medicine and Pharmacy Cluj-Napoca; registration number, 50; date, 11 March 2019) and written informed consent was provided by all subjects, owing to its retrospective nature. The database consisted of our radiology information system (RIS). A keyword search allowed us to select 61 patients presenting intraperitoneal collections visible on abdominopelvic MRI examinations carried out between March 2016 and April 2019.

The inclusion criteria were: a final diagnosis regarding the intraperitoneal collections based on cytological and histopathological findings, patients who underwent MRI less than one month before or after fluid analysis, subjects with no more than one pathology that could cause intraperitoneal fluid to accumulation, and patients whose pathological analysis of the fluid did not detect major contamination of the collections (especially with blood).

According to the cytological result of the fluid analysis (e.g., the presence or the absence of malignant cells in the sampled fluid), subjects were included in the malignant collections group (n = 24) and non-neoplastic (benign) collections group (n = 37). Paracentesis was performed in 29 patients, laparoscopies were performed 20, and laparotomies in 12 patients. The flow diagram is displayed in Figure 1.

Twenty-two patients had a final diagnosis of cirrhosis (virus B-related cirrhosis, n = 8; virus C-related cirrhosis, n = 5; alcoholic cirrhosis, n = 8; primitive biliary cirrhosis, n = 1). Intra-abdominal abscesses (or secondary peritonitis) were due to acute appendicitis in two subjects and, respectively, Crohn’s disease and postoperative abscess, each in one subject. Nephrogenic ascites was caused by focal segmental glomerulosclerosis and membranous nephropathy, each in one subject. Pancreatitis-related collections were due to pancreatic ascites in three patients, infected peripancreatic fluid collections in one and pseudocysts in two subjects. From the malignant group, six patients were demonstrated with serous and four with clear cell ovarian carcinomas. The presence of malignant cells was detected in every fluid sample of the subjects comprised in the malignant ascites group. Details about the pathologies included in each group are shown in Table 1.

### 2.2. Fluid Analysis

All fluid samples were processed by the same laboratory. The sampled fluids were divided for cytological and biochemical analysis and additional ancillary tests. The mean volume of the sampled fluids was 5.7 mL (range 2–15 mL). For cytological analysis, the probes were first centrifuged. From each probe, two pellets were assembled, stained with hematoxylin and eosin and analyzed microscopically. The four abscesses, as well as the two pseudocysts and the infected peripancreatic fluid collections, were surgically removed and underwent pathological analysis. Since their underlying disease was well documented, the diagnosis of these lesions was straightforward and were described by pathologists by their gross appearance. Sixteen patients underwent multiple liquid sampling procedures for reasons such as suspected fluid infection, or evacuation of a symptomatic collection. Thirty-two patients underwent laparotomies or laparoscopies after the MRI examination, and in three cases (two ovarian carcinomas and one pancreatic ascites) the intra-peritoneal collections were resampled in the evolution of their disease (mean time from surgery to resample, 167 days; range 125–219 days).

### 2.3. MRI Examination and Image Interpretation

All abdominopelvic MRI scans were performed on the same unit (General Electric Optima 360MR Advance system, Waukesha, WI, USA) with dedicated array coils, covering abdominal and pelvic region: GEM (Geometry Embracing Method) Suite. Patients were informed about the duration of the examination, were instructed not to move, and to maintain expiratory apnea according to instructions and breathe constantly for respiratory triggered sequences. Headphones were used for communication and ear protection.

Abdominal exploration summarizes the following standard sequences: axial T1 GRE Dual Echo BH (T1 Gradient Echo In/Out of Phase Breath Hold), axial T2 SS FSE BH (T2 Single Shot Fast Spin Echo Breath Hold) and FIESTA (Fast Imaging Employing Steady-state Acquisition), coronal T2 FatSat SS FSE BH and FIESTA. An extended field of view (FOV) was used, adapted to the size of the patients (FOV = 38–48 cm). The number of slices was set to be sufficient to cover the area between the diaphragm and iliac crests. Axial DWI, synchronized with respiratory movements (RTr), with 4 b values (50, 200, 600, and 800 s/mm^2^), was performed using the same slice thickness, interval, and location as for standard axial sequences. Imaging parameters: repetition time (TR), 10,000 ms; echo time (TE), 64 ms; slice thickness, 6 mm; interval, 1 mm and acquisition matrix, 128 × 128. The number of averages for b values were 2, 3, 5 and 8, respectively. The pelvic examination included: axial T1 FSE (Fast Spin Echo) whole pelvis with large FOV, sagittal, axial and coronal T2 Periodically Rotated Overlapping Parallel Lines with Enhanced Reconstruction (PROPELLER) with an FOV of 26–30 cm. Axial DWI echo-planar imaging (EPI), with 3 b values (50, 400, and 1000 s/mm^2^) was set using the same slice thickness, interval, and location as the standard axial T1 FSE sequence. The DWI parameters were: TR, 4000 ms; TE, 94 ms; slice thickness, 6 mm; range, 1 mm; matrix, 128 × 128. The number of averages for b values were 1, 5 and 10. The ADC and Exponential Apparent Diffusion Coefficient (eADC) functional maps were automatically obtained on the scanner computer, using EPI Correction and following the same parameters, for both abdominal and pelvic acquisition: Lower Threshold, 20; Confidence Level, 0.9; Kernel Size, 2.

Quantitative analyses were performed on a dedicated workstation (General Electric, Advantage workstation, 4.7 edition) by two radiologists (CC and AL, each with at least 15 years’ experience in abdominal imaging), blinded to the final diagnosis. Each radiologist measured the fluid from at least two sites (perihepatic and perisplenic), by placing an elliptical or spherical region of interest (ROI) of two unlike locations from each site. When the ascites fluid was visible only in one region, three ROIs were placed on separate locations on the selected region. The minimum area of each ROI was set at 10 mm^2^. The ROI was placed on the slice with the most ascites, while maintaining a distance of at least three quarters diameter of the ROI relative to the surrounding tissues. All ROIs have been drawn on the ADC maps with reference to the DWI and T2 FatSat FIESTA sequences to ensure that it is not placed on solid structures or artifacts (Figure 2). The values were averaged separately in order to obtain the mean ADC values for each patient, which were subsequently used to compare selected groups.

### 2.4. Statistical Analysis

The normality of the data distribution was analyzed consuming the Kolmogorov–Smirnov test. The independent samples *t*-test was conducted to test for differences between the mean ADC values of the two groups. A *p* value of less than 0.01 was considered significant after Bonferroni correction. The same test was consumed to investigate the difference between ADC values measured on the abdomen and pelvis regions. A receiver operating characteristic (ROC) analysis was used to evaluate the diagnostic power of ADC values in differentiating malignant from benign IPCs, and the sensibility and specificity were showed for an optimal cut-off value. An inter-rater agreement (Kappa) test was conducted in order to evaluate the agreement between the values measured by the two radiologists. Statistical analysis was performed using a commercially available dedicated software MedCalc version 14.8.1 (MedCalc Software, Mariakerke, Belgium).

## 3. Results

Sixty-one patients were retrospectively included in this study (27 females and 34 males; mean age 61.08, age range 34–89 years). The mean time from the fluid sampling to the MRI examination was 8.7 days (range, 2–29 days). The ROIs were placed on perihepatic and perisplenic collections in 44 subjects and on a single location in 17 patients. Thirty-seven ROIs were placed in the abdomen and 24 in the pelvic region. From the benign group, 15 ROIs were placed in the pelvis and 22 in the abdomen, while for malignant group 9, ROIs were placed in the pelvis and 15 in the abdomen. The mean ADC values measured from abdominal regions were 3.384 × 10^−3^ mm^2^/s (standard deviation, 0.558 × 10^−3^ mm^2^/s) and 3.216 × 10^−3^ mm^2^/s (standard deviation, 0.637 × 10^−3^ mm^2^/s) from the pelvic region. The comparison of the measurements made in the two locations were not statistically significant (*p* = 0.0818). The average ROI size was 17.7 mm^2^ (range, 10.6–37.3 mm^2^). The inter-rater agreement test resulted in a K value of 0.44, which qualifies for a moderate strength of agreement between de ADC values measured by the two radiologists.

The Kolmogorov–Smirnov test for normality resulted in a *p* value of 0.81 for benign and 0.19 for malignant IPCs measurements, which indicated that in both cases the data are normally distributed. The mean ADC values were 3.543 × 10^−3^ mm^2^/s (range, 2.363–4.846 × 10^−3^ mm^2^/s; standard deviation, 0.540) for the benign group and 3.057 × 10^−3^ mm^2^/s (range, 2.47–4.42 × 10^−3^ mm^2^/s; standard deviation, 0.574) for the malignant group (Figure 3), and the difference between the ADC values measured for each group was statistically significant (*p* = 0.0034). The optimum cut-off point of the ADC values in the diagnosis of malignant ascites was <3.241 × 10^−3^ mm^2^/s, with a sensitivity of 77.78% (52.4–93.6, 95% confidence interval (CI)) and a specificity of 80% (64.4–90.9, 95% CI). The area under the curve was 0.734 (0.62–0.841, 95% CI), the Youden index (J) was 0.57, and the significance level (P) was 0.005 (Figure 4).

## 4. Discussion

Peritoneal carcinomatosis is a late stage manifestation of several malignancies characterized by tumor deposition across the peritoneal surface [12]. Although MRI is a useful diagnostic tool in assessing the peritoneal changes advocating for malignancy, it is often limited by the high cost and long imaging times [13]. Furthermore, if the peritoneal implants are placed between intraluminal air and mesenteric fat or in the mesentery, the administration of specific contrast agents can cause some obscuration between the lesion and fat [14]. On the other hand, MRI incorporates a functional technique known as DWI, which provides information about the Brownian motion of water molecules in a tissue [15]. In recent years, this technique has emerged as a new method for the characterization of fluids at a molecular level [16]. The ADC values represent a quantitative measurement of the degree of such motion in tissue, and are also used as a marker of cellularity [15]. Aside from being a noninvasive, reliable, and reproducible imaging parameter, the ADC measurements have been proven to be useful in the evaluation and characterization of different types of effusion [15,17]. To our knowledge, research involving these methods has been limited to the pleural fluid to date, and successfully proved that differences in liquid content have an important influence on ADC values [16,17,18].

The b value is a factor that reflects the strength and timing of the gradients used to generate diffusion-weighted images [19]. The DWI images are created by diffusion-sensitizing gradients turned on at various strengths [20]. The b value is directly linked to the diffusion effects [19]. The ADC maps in our study were automatically computed using all b values for pelvic and abdominal examinations, respectively.

The structural components of the tissue, as well as the microcirculation of blood in the capillary network (perfusion), influence the microscopic motion detected by DWI [21]. Different choices of b values for the computation of ADC maps can lead to variations in the resulting absolute ADC values [22], as well as in lesion delineation and visual ADC contrast [23]. Lesion delineation, together with visual contrast, were not particularly important in the current research, since our work involved targeting large fluid areas and not small lesions within the organs. What may have impacted the results of our study is the use of two different ADC maps computed from two sets of b values (50, 200, 600, and 800 s/mm^2^ for the abdominal and 50, 400, and 1000 s/mm^2^ for the pelvic maps). Thörmer et al. [24] demonstrated that the ADC values were inversely correlated with the b values used for the ADC calculation. This effect may be caused by an increased contribution of tissue perfusion and motion at smaller b values [24]. However, the influence of tissue perfusion on the values measured on our ADC maps was null since we did not target an organ, but a fluid that has no vascularization. There are contradictory reports regarding the higher b-values’ contribution to increasing the absolute ADC values [23,25,26]. We were unable to find a study that compared the exact values that we used for ADC maps computation. Our results showed higher average ADC values extracted from the abdomen than from the pelvis maps, but the difference between the two was not statistically significant (*p* = 0.0818).

We are aware that the ADC values can vary even within the same examination. In this regard, we conducted a workflow where at least six-to-eight measurements were made from the separate regions of each collection. Moreover, these measurements were averaged and only the resulting value was used for statistical analysis. Through this method, we aimed to counteract, at least mathematically, part of the ADC variations that could be encountered due to sedimentation or the use of multiple b values.

The dimension of the sampled area can also influence the ADC values. However, this influence is highly dependent on the type of lesion and organ. While Zhou et al. [27] observed that small ROIs have a negative effect on ADC differentiation of benign from cancerous thyroid nodules, Gity and colleagues [28] noted that the reduced ROI area can augment the diagnosis of benign versus malignant tumors in mass and non-mass breast lesions.

The choice of using multiple ROIs instead of incorporating collections in a larger volume of interest (VOI) could have influenced the results of our study. Miquel et al. [29] observed ADC fluctuations between slices of examinations comprising abdominal organs and concluded that these variations are less likely to affect three-dimensional VOIs because any differences between (and within) slices are likely to be averaged over the large VOI. Unfortunately, less in-slice averaging and no interslice averaging is present when using smaller two-dimensional ROIs [29]. However, the same research [29] reported higher coefficients of reproducibility for the ROI than for the VOI analysis. We agree that the use of VOIs would have provided a more accurate description of the diffusion within the collections, overcoming regional fluctuations. Our method, although less accurate, is closer to the actual ADC measurement methodology applied in clinical practice, straightforward and less time-consuming. We believe that, if validated in larger prospective studies, this assessment technique will use a similar method, since manual or semi-automatic VOI delineation will require longer segmentation times.

On the other hand, we successfully counteracted inter-scanner variabilities in ADC measurements and the effect of different post-processing software [30] on ADC values by selecting only examinations that were performed on the same machine and processed on the same workstation. Previous research obtained up to 4% variability when using different MRI machines [31] and near 8% variation when ADC was processed on different types of software [32].

In pathological analysis, malignant ascites present specific characteristics, such as high concentrations of proteins and cholesterol [33] and an increase in lactate dehydrogenase [34]. Mesothelial cells (77.77%) and erythrocytes (59.25%) represented the two major types of cells [33], along with leukocytes [34]. Since intra-abdominal tumors are one of the most common causes of chylous ascites in adults, on gross examination, the liquid can appear milky due to the presence of chylomicrons [8]. In contrast, nonmalignant peritoneal fluid may occur in many forms [35,36]. On gross examination, the fluid may appear clear (in liver cirrhosis) or cloudy (in bacterial peritonitis, perforated bowel, and pancreatitis) [35]. It can also have a turbid appearance due to the presence of neutrophils (when it is associated with bacterial peritonitis and pancreatitis) and triglycerides (in cirrhotic patients), being hard to differentiate from true chylous ascites [35,36].

The ADC values were statistically significant when comparing malignant and non-neoplastic collections (*p* = 0.0034). The average values were higher for the benign than for the malignant group. Two hypotheses can be formulated regarding this difference. The first refers to the high protein content that is characteristic of neoplastic ascites and its increased viscosity due to the chylomicron content, both of which can lead to a decrease in ADC values [7,10]. It is, therefore, possible that the measurements strictly reflect the differences in the physical properties of the two liquids. On the other hand, most of the patients in the benign group had inflammatory-infectious collections, which are also characterized by high cellularity and increased density [6] that can also limit the movements of water molecules [11]. Thus, it is possible that the differences in ADC values between the two entities may reflect more than the physical characteristics of the liquids, being the result of the distinct histopathological features (such as the presence of neoplastic cells organized in clusters).

There is a common misconception that malignancy-related ascites (MRA) is synonymous with peritoneal carcinomatosis [37]. MRA signifies fluid accumulation in the peritoneum which appears in the context of neoplastic diseases and can have multiple causes [7]. In approximatively one-third of oncological patients, ascites are due to altered vascular permeability and obstructed lymphatic drainage, and therefore are not associated with the presence of tumor cells in the fluid [7,37]. On the other hand, neoplastic ascites (NA) indicate the presence of malignant cells in the peritoneal cavity [7] and can be positive only in patients who have cancerous cells lining the peritoneum with the shedding of viable cells into the fluid collections [38]. A comparison between the ADC values obtained from MRA with negative cytology and NA would be beneficial to the more accurate observation of the influence that the presence of neoplastic cells has on the diffusion of water molecules. Although several cases in which intra-abdominal tumors (especially liver and pancreatic) were associated with ascites fluid with benign cytology were identified in our database, they were not included in the study because another cause of fluid accumulation (such as portal hypertension) could also have been involved.

The malignant etiology of ascites can be diagnosed by positive cytology with a specificity of up to 100% [39]. However, the sensitivity of this method in detecting malignant aspirates within PC is variable. The best results were achieved by Runyon et al. [38], which showed that 96.7% of the patients with peritoneal carcinomatosis, had positive or suspicious cellularity. This good result was probably due to the elaborate protocol that the authors used for the analysis: the fluid was sampled only by paracentesis, initial negative results were reevaluated with a sampling of another larger specimen, and the timing of paracentesis and handling of probes was coordinated with the laboratory [38]. This scenario model was not applied in our study. We used multiple sample techniques, initial negative results were not reevaluated in the same hospitalization period and there was no coordination between the timing of the procedure and the laboratory. Considering this, it is safe to assume that the sensibility in detecting malignant cells was lower in our research. Two large retrospective studies [40,41] that retrieved information about fluid samples analyzed under standard healthcare unit protocols showed that cytology was able to detect malignant aspirates with a sensitivity of 57% [40], and 60%, respectively [41]. The methodologies used in these studies [40,41] seem to be more closely related to our workflow. In addition to these observations, various factors could influence the ability of cytological analysis to detect malignant cells. The best-suited fluid for this analysis is the one in its natural state and without added preservatives. Secondly, the inadequate sampling and transportation to the laboratory can lead to degeneration or compromise the viability of the cells [42]. Other substances mixed with the fluid could affect the diagnostic accuracy (such as heparin influencing the pH measurements) [43]. Since the fluid sampling and processing techniques are not regulated by protocols in our healthcare unit, these methods rely on the knowledge and experience of the examiners. We were unable to retrieve any data referring to the sampling maneuvers’ workflow and fluid transportation from the medical records of the patients included in this study. For intra-peritoneal collections, a minimum volume of approximately 100 mL is recommended to ensure the adequacy of the sample for proper cytological processing and evaluation [43]. Unfortunately, our analyzed fluid samples were almost ten times lower.

Our study had several limitations. First, owing to its retrospective design, it could have selection bias. The cytological analysis was the only criterion that decided the distribution of patients in the two groups without considering other liquid characteristics. However, the cytological criterion is not pathognomonic for the diagnosis of malignant cells within intra-peritoneal collections, and the sensitivity of this analysis could have also been decreased by the sampling and processing methods. The biochemical and physical fluid features, although they could theoretically influence the intensity of the MRI signal, are extremely variable even within the same pathology, and their investigation based on the ADC maps would be of less importance in the clinical practice. Unfortunately, we could not find any correlations between ADC values and the biochemical features of the fluids, since the latter were not available for all subjects.

## 5. Conclusions

Our research has shown that there are statistically significant differences between the two types of intraperitoneal collections, which can be attributed to the physical characteristics of the fluids, specific cellularity or both. Being the first research to investigate the difference between benign and malignant intraperitoneal collections based on diffusion and ADC maps, the results are promising. The study opens the way for future research designed to validate this method and to accurately identify the dynamics of ADC values based on ascites fluid components and properties. If confirmed on a larger number of patients, this approach may have an important role in the non-invasive diagnosis of ascites, with multiple benefits in current practice.

## Figures and Tables

**Figure 1 medicina-56-00217-f001:**
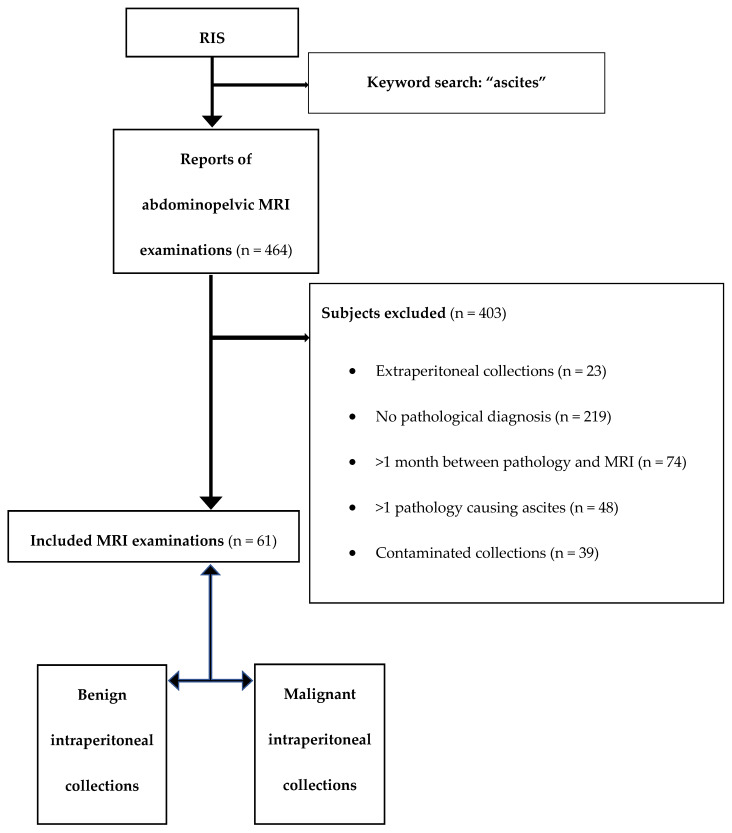
Flow diagram. RIS, radiology and information system; MRI, magnetic resonance imaging.

**Figure 2 medicina-56-00217-f002:**
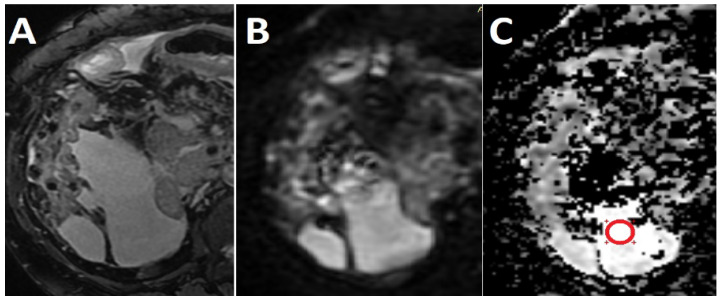
Ascites in a 53-year old patient with alcoholic liver cirrhosis. Axial T2 FatSat Fast Imaging Employing Steady-state Acquisition (FIESTA) (**A**) and Axial diffusion-weighted imaging (DWI) computed using b = 50 s/mm^2^ (**B**) sequences used as guidance for placing a circular region of interest (ROI) on the apparent diffusion coefficient (ADC) map (**C**).

**Figure 3 medicina-56-00217-f003:**
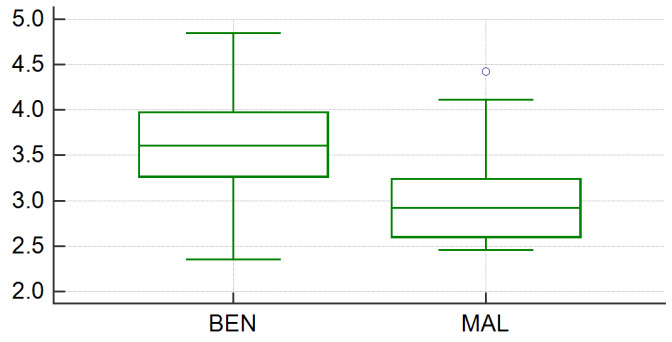
Box plot using bars to show mean ADC values for benign (BEN) and for malignant (MAL) intraperitoneal collections group.

**Figure 4 medicina-56-00217-f004:**
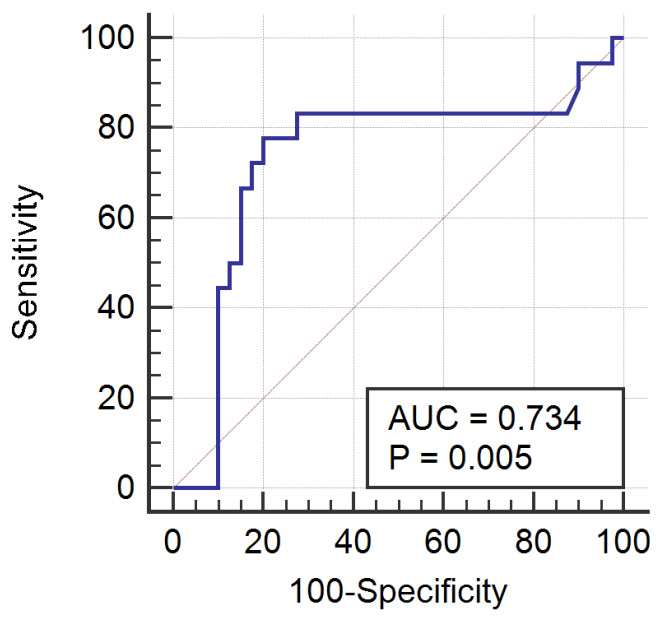
The receiver operating characteristic curve of ADC values for the differentiation between benign and malignant intraperitoneal collections. AUC, area under the curve. P, significance level.

**Table 1 medicina-56-00217-t001:** Patient groups.

**Benign Group** (n = 37)	**n**
liver cirrhosis	22
peritoneal tuberculosis	1
intra-peritoneal abscess	4
intestinal ischemia	2
nephrogenic ascites	2
pancreatitis-related collections	6
**Malignant Group** (n = 24)	
colorectal cancer	7
gastric cancer	5
endometrial carcinoma	2
ovarian carcinoma	10

n, number of patients.

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
