# Peer review of "Diffusion-Weighted Magnetic Resonance Imaging as a Noninvasive Parameter for Differentiating Benign and Malignant Intraperitoneal Collections"

_medicina, 2020, doi:10.3390/medicina56050217_

Round 1

Reviewer 1 Report

The present retrospective study assesses the potential of diffusion-weighted magnetic resonance imaging (DW-MRI) in distinguishing between benign and malignant ascites.
The study is clinically relevant since the differentiation of benign and malignant ascites is essential for treatment stratification and monitoring of patients. The sensitivity and specificity of contrast enhanced CT and MRI for the detection of peritoneal metastases is comparatively low. This urges for the establishment of non-invasive methods for detection of peritoneal carcinomatosis/ malignant ascites.

Thematically, the manuscript is suited for a journal focussing on clinical investigations, like Medicina.

However, some major points should be addressed before publication.

Introduction:
The introduction is well written. It briefly and precisely explains the background of the study and its aims.

Materials and Methods and Results:
1) The authors included 61 patients with intraperitoneal fluid collections. Ascitic cytology served as gold standard for assignment of patients to a "benign ascites group" and a "malignant ascites group". The reported sensitivities of ascitic cytology for the detection of peritoneal carcinomatosis are highly variable in the literature and probably dependent on the exact methods of the cytologic analysis. The authors cite a study by Runyon et al. ( https://doi.org/10.1002/hep.1840080521) which reports a sensitivity of 96.7%. Their workup of the ascites fluid specimens was rather sophisticated (If malignancy was strongly suspected despite an initial negative ascitic fluid cytology, a second large (1,000 ml) specimen was analyzed; timing of the paracentesis and the handling of the cytology specimen were coordinated with the cytology laboratory), which might explain the high sensitivity value in the study by Runyon et al. In other studies, the sensitivity of ascitic cytology for the diagnosis of peritoneal carcinomatosis was below 80% (https://www.ncbi.nlm.nih.gov/pmc/articles/PMC1731306/).
Therefore, the authors of the present study should further specify their methods of ascitic fluid analysis.

2) The "benign ascites group" includes, among others, patients with intra-abdominal abscesses, infected pancreatic fluid collections and pseudocysts. Technically speaking, abscesses and pseudocysts are not ascites.

3) The authors state that ADC maps and eADC maps were obtained. Were only the ADC maps used for further analysis? Where the ADC maps calculated from all b values?
The b values differed between abdominal and pelvic MRI scans. The choice of b values can influence the ADC (https://doi.org/10.1007/s00330-012-2432-3). This potential confounder should be discussed by the authors. How many patients from the "benign ascites group"/ "malignant ascites group" had pelvic MRIs/ abdominal MRIs, each? The units of b values (s/mm²) should be added to the Materials and Methods part.

4) "The minimum diameter of each ROI was set at 10 mm2." Is it diameter or area?
The placement of small ROIs for the determination of ADC values of ascites is probably not sufficient. In my experience, ADC values of the ascitic fluid vary greatly within the same examination, possibly due to sedimentation effects. The ADC map in Figure 1C shows that the ADC values vary between the area of the ascitic fluid where the ROI is placed and the further dorsal part of the ascitic fluid which is darker (indicating lower ADC values). I would recommend using an image analysis software (e.g. MITK diffusion, http://www.mitk.org/wiki/MitkDiffusion) which allows for placement of volumes of interest (VOIs) rather than regions of interest (ROIs) to better account for the variability of ADC values within different areas of ascitic fluid.
The b value of the diffusion weighted image in Figure 1B should be added to the figure legend.

Discussion:
1) "Peritoneal carcinomatosis is defined as intraperitoneal dissemination of any cancer, apart from primary peritoneal tumors." This sentence is not correct. For instance, disseminated intraperitoneal spread of a sarcoma is not called "peritoneal carcinomatosis", but "peritoneal sarcomatosis".

2) The authors should consider further discussing the limitation that cytology alone served as gold standard for assignment of patients to the "benign ascites group" and the "malignant ascites group".

Author Response

Dear Reviewer,

Thank you for your work in reviewing this manuscript and for your interest in our research.

In the following we will detail the changes made to the study, according to your suggestions. We hope the adjustments we made meet your expectations.

Materials and Methods and Results:

1) The authors included 61 patients with intraperitoneal fluid collections. Ascitic cytology served as gold standard for assignment of patients to a "benign ascites group" and a "malignant ascites group". The reported sensitivities of ascitic cytology for the detection of peritoneal carcinomatosis are highly variable in the literature and probably dependent on the exact methods of the cytologic analysis. The authors cite a study by Runyon et al. ( https://doi.org/10.1002/hep.1840080521) which reports a sensitivity of 96.7%. Their workup of the ascites fluid specimens was rather sophisticated (If malignancy was strongly suspected despite an initial negative ascitic fluid cytology, a second large (1,000 ml) specimen was analyzed; timing of the paracentesis and the handling of the cytology specimen were coordinated with the cytology laboratory), which might explain the high sensitivity value in the study by Runyon et al. In other studies, the sensitivity of ascitic cytology for the diagnosis of peritoneal carcinomatosis was below 80% (https://www.ncbi.nlm.nih.gov/pmc/articles/PMC1731306/).

Therefore, the authors of the present study should further specify their methods of ascitic fluid analysis.

Answers:

- We addressed the fluid analysis methodology followed by Runyon et al. (https://doi.org/10.1002/hep.1840080521) in comparison to more standard protocols and to ours in a separate paragraph at the end of the “Discussion” section.

- The fluid analysis protocol has been added to the “Materials and method” section. (page 3, lines 119-132).

2) The "benign ascites group" includes, among others, patients with intra-abdominal abscesses, infected pancreatic fluid collections and pseudocysts. Technically speaking, abscesses and pseudocysts are not ascites.

Response: Thank you for the observation. When necessary, the terms “benign /+ ascites” have been replaced with “intraperitoneal /+ collections (IPCs)”.

3) The authors state that ADC maps and eADC maps were obtained. Were only the ADC maps used for further analysis? Where the ADC maps calculated from all b values? The b values differed between abdominal and pelvic MRI scans. The choice of b values can influence the ADC (https://doi.org/10.1007/s00330-012-2432-3). This potential confounder should be discussed by the authors. How many patients from the "benign ascites group"/ "malignant ascites group" had pelvic MRIs/ abdominal MRIs, each? The units of b values (s/mm²) should be added to the Materials and Methods part.

Answers:

- Yes, only the ADC maps used for further analysis.

- Yes, the ADC maps were calculated automatically including all b values. We added a statement in the “Discussion” section where we also addressed other matters referring to ADC measurements (page 7, line 239: “The ADC maps in our study were automatically computed using all b values for pelvic and abdominal examinations, respectively.”)

- The influence of different b values on the ADC measurements in this study are now referred extensively in the “Discussion” section. We also addressed the variability of ADC measurements within the same examination, the use of multiple ROIs, the use of VOIs (as you mentioned later in the review) as well as other factors that influence the ADC values such as scanners and post-processing software.

- The use of both pelvic and abdominal MRI maps was addressed in the “Results” and “Discussion” section. From the benign group 15 ROIs were placed in the pelvis and 22 in the abdomen, while for malignant group 9 ROIs were placed in the pelvis and 15 in the abdomen. We also conducted a univariate analysis test to check if there was a statistically difference between the two measurements, and the result (P=0.08) showed there isn’t one.

- The units of b values were added to the “MRI examination and image interpretation” section within Materials and methods. (page 4, lines 145-6; 152).

4) "The minimum diameter of each ROI was set at 10 mm2." Is it diameter or area?

The placement of small ROIs for the determination of ADC values of ascites is probably not sufficient. In my experience, ADC values of the ascitic fluid vary greatly within the same examination, possibly due to sedimentation effects. The ADC map in Figure 1C shows that the ADC values vary between the area of the ascitic fluid where the ROI is placed and the further dorsal part of the ascitic fluid which is darker (indicating lower ADC values). I would recommend using an image analysis software (e.g. MITK diffusion, http://www.mitk.org/wiki/MitkDiffusion) which allows for placement of volumes of interest (VOIs) rather than regions of interest (ROIs) to better account for the variability of ADC values within different areas of ascitic fluid.

The b value of the diffusion weighted image in Figure 1B should be added to the figure legend.

Responses:

- Thank you for the observation. The correct term is “area”. We made the necessary adjustment.

- We addressed the use of multiple ROIs and VOI use in the first part of the “Discussion” section, along with other factors that influence the ADC measurements (pages 7,8 & 9: lines 236-288). We also acknowledged that the method suggested by you could provide more accurate scientific results. Through this study, we aimed to evaluate the difference between the two entities through an approach that resembles more the clinical practice (2D ROI placement). We consider your suggestion for future research; we have already set the basis for a larger prospective study that will assess different types of tissues (including collection) by using textural information extracted by 3D ROIs.

- The b value for the DWI image was added in figure description, it now states: “Axial DWI image at b=50 s/mm2

Discussion:

1) "Peritoneal carcinomatosis is defined as intraperitoneal dissemination of any cancer, apart from primary peritoneal tumors." This sentence is not correct. For instance, disseminated intraperitoneal spread of a sarcoma is not called "peritoneal carcinomatosis", but "peritoneal sarcomatosis".

Response: Thank you for the observation. The affirmation was replaced with: “Peritoneal Carcinomatosis (PC) is a late stage manifestation of several malignancies characterized by tumor deposition across the peritoneal surface”

2) The authors should consider further discussing the limitation that cytology alone served as gold standard for assignment of patients to the "benign ascites group" and the "malignant ascites group".

Response: This matter is now addressed in a separate paragraph in the “Discussion” section (pages 10&11: lines 325-51).. We acknowledge that are pitfalls regarding the cytological analysis of the fluids we selected. In this regard we also included an affirmation in the “study limitations” paragraph (“But the cytological criterion is not pathognomonic for the diagnosis of malignant cells within intra-peritoneal collections, and the sensitivity of this analysis could have also been decreased by the sampling and processing methods of the probes included in our study”).

We wish to warmly thank you for expert, thoughtful and very pertinent observations, which made us realize omissions we made and, we estimate, greatly helped us improve the paper.

With gratitude,

Csutak Csaba & Paul Stefan

Reviewer 2 Report

This manuscript evaluated ascitic fluid with an apparent diffusion coefficient (ADC) to determine non-invasive differentiation criteria for benign and malignant intraperitoneal effusions in 61 patients. The study validated the method to accurately identify the dynamics of ADC values based on ascites fluid components and properties. Though it would be very useful to include more number of subjects, however, the current study provides important information on the role of the non-invasive diagnosis of ascites using Diffusion-Weighted Magnetic Resonance Imaging (DWI). The introduction and methods were appropriate but the results and discussion sections should be improved further with the following suggestions:  

Major comments

  1. The most important outcome of this study is the use of Apparent Diffusion Coefficient (ADC) values for distinguishing malignant and benign and malignant intraperitoneal effusions. Literature indicates that a number of factors such as measuring methods and choice of b values influence the ADC values and thus, question the risk to use the reported ADC thresholds in clinical practice. It would be important and useful to perform a meta-analysis on the data with a discussion of factors influencing the ADC values.

  1. Though authors provided the receiver operating characteristic curve of ADC values with sensitivity and specificity but they have not discussed the heterogeneity. Perhaps it would be worth calculating the heterogeneity with the available methods for diagnostic test accuracy such as inconsistency index I2.

  1. It would be important to include a flow diagram for data acquisition for study. This will be helpful to judge bias.

  1. Page 6: line 218: The statement ‘Also, the cytological analysis of the ascitic fluid was the only criterion that decided the distribution of patients in the two groups without considering other liquid characteristics.’ Since the cytological analysis was the only criterion for the patient distribution, it would be very informative to correlate the pathology with available T2-weighted images.

Minor comments

  1. Page 2: line 77, ‘Paracentesis was performed in 29 patients, laparoscopy was performed 20, and laparotomy in 12 patients’. In this sentence, singular nouns ‘laparoscopy’ and ‘laparotomy’ follow a number other than one. Consider changing the nouns to the plural forms and correct the sentence.
  2. Page 4: Figure 2 caption, ‘The receiver operating characteristic curve of ADC values for the diferentiation between bening and malignant ascites.’ In this sentence, ‘diferentiation’ and ‘bening’ should read as ‘differentiation’ and ‘benign’ respectively. Please consider correcting the typos.
  3. Page 6: line 215: ‘they were not included in the study because another cause of fluid accumulation (such as portal hypertension) could have also been involved.’ It appears that there is a missing preposition after the word ‘because’. Consider adding a preposition.
  4. The values reported in the manuscript do not include the correct notation for cube and square. Please consider superscript the values wherever required throughout the manuscript. For example, Page 10, line 121: ‘10mm2’ should be ‘10 mm2

Author Response

Dear Reviewer,

Thank you for your work in reviewing this manuscript and for your interest in our research.

In the following, we will detail the changes made to the study, based on your suggestions. We hope the adjustments meet your expectations.

Major comments

The most important outcome of this study is the use of Apparent Diffusion Coefficient (ADC) values for distinguishing malignant and benign and malignant intraperitoneal effusions. Literature indicates that a number of factors such as measuring methods and choice of b values influence the ADC values and thus, question the risk to use the reported ADC thresholds in clinical practice. It would be important and useful to perform a meta-analysis on the data with a discussion of factors influencing the ADC values.

Response: Our study design was experimental and retrospective, thus we were able to integrate a succinct analysis regarding the factors influencing the ADC values and how they affected the workflow and outcome of our study. In this analysis we addressed the variability of ADC measurements by the use of multiple b values, variabilities within the same examination, the use of multiple ROIs, how are ADC values affected by the use of 2D and 3D regions of interest, as well as external factors such as scanners and post-processing software. This analysis was added to the “Discussions” section (pages 7,8 &9: lines 236-287).

Though authors provided the receiver operating characteristic curve of ADC values with sensitivity and specificity but they have not discussed the heterogeneity. Perhaps it would be worth calculating the heterogeneity with the available methods for diagnostic test accuracy such as inconsistency index I2.

We searched for a method to assess the heterogeneity of the data included in our study. It seems that the I2 test is able to show heterogeneity, but only in observational studies (e.g. meta-analysis, 10.1136/bmj.327.7414.557). Similarly, another test that investigates whether observed differences in results are compatible with chance alone, such as chi-squared (χ2, or Chi2), is also suited only for meta-analysis. Although we have limited experience in the field of medical statistics, we could not find a statistical test that could investigate the heterogeneity of data in experimental studies, and we would be very grateful if you could suggest a method to test the heterogeneity of the data. Meanwhile, we will continue to investigate this issue.

It would be important to include a flow diagram for data acquisition for study. This will be helpful to judge bias.

 Response: The flow diagram is now included as “Figure 1”.

Page 6: line 218: The statement ‘Also, the cytological analysis of the ascitic fluid was the only criterion that decided the distribution of patients in the two groups without considering other liquid characteristics.’ Since the cytological analysis was the only criterion for the patient distribution, it would be very informative to correlate the pathology with available T2-weighted images.

Response: We acknowledge some of the pitfalls regarding the use of cytological analysis as the gold standard for discriminating between the two groups in our study. These limitations have been addressed in a separate paragraph at the end of the “Discussion” section (pages 10&11: lines 325-51). Our work focused on the distinction of the two fluid types based on ADC maps since we theorized that the difference in cellularity will cause modifications in the diffusion of water molecules. Since the tissues we targeted are mostly composed of water, we were unable to do this analysis on the T2-weighted sequence, mostly because both types of fluids express high signal intensities, and the sequence does not carry information about water diffusion. Also, the signal intensity measurements are not routinely used on standard MRI sequences. However, it is possible that these sequences can carry additional diagnostic information. In this regard, we already started working on a study that involves the assessment of classic MRI sequences by the use of texture analysis, on different types of tissues (including fluids).

Minor comments

Page 2: line 77, ‘Paracentesis was performed in 29 patients, laparoscopy was performed 20, and laparotomy in 12 patients’. In this sentence, singular nouns ‘laparoscopy’ and ‘laparotomy’ follow a number other than one. Consider changing the nouns to the plural forms and correct the sentence.

Response: Modifications were made according to suggestion.

Page 4: Figure 2 caption, ‘The receiver operating characteristic curve of ADC values for the diferentiation between bening and malignant ascites.’ In this sentence, ‘diferentiation’ and ‘bening’ should read as ‘differentiation’ and ‘benign’ respectively. Please consider correcting the typos.

Response: The Grammarly was corrected. 

Page 6: line 215: ‘they were not included in the study because another cause of fluid accumulation (such as portal hypertension) could have also been involved.’ It appears that there is a missing preposition after the word ‘because’. Consider adding a preposition.

Response: Yes, we added the preposition “of”. It now states: “…they were not included in the study because of another cause of fluid accumulation…”

The values reported in the manuscript do not include the correct notation for cube and square. Please consider superscript the values wherever required throughout the manuscript. For example, Page 10, line 121: ‘10mm2’ should be ‘10 mm2’

Reponses: Changes were made according to suggestions.

We wish to warmly thank you for expert, thoughtful and very pertinent observations, which made us realize omissions we made and, we estimate, greatly helped us improve the paper.

With gratitude,

Csutak Csaba & Paul Stefan

Round 2

Reviewer 1 Report

The authors have well revised the manuscript according to the reviewer's suggestions. The exact method of fluid analysis is now decribed in detail. Also, the limitations that result from the use of fluid analysis as gold standard for the differentiation of malignant and benign collections are extensively discussed which is very good.

I would consider replacing the term "types of ascites" in the first sentence of the Conclusions part (line 365) with "intraperitoneal collections".

Author Response

Dear Reviewer,

Thank you for taking the time to analyze our manuscript. Also, we are grateful for your help in improving this paper. We have made the suggested changes (page 12, line 365: the term “ascites” has been replaced with “intraperitoneal collections”).